# Distribution Pattern, Emission Characteristics and Environmental Impact of Polycyclic Aromatic Hydrocarbons (PAHs) in Download Ash and Dust from Iron and Steel Enterprise

**DOI:** 10.3390/molecules24203646

**Published:** 2019-10-09

**Authors:** Youmin Sun, Chunzhu Chen, Chun Ding, Guanghui Liu, Guiqin Zhang

**Affiliations:** 1School of Municipal and Environmental Engineering, Shandong Jianzhu University, Jinan 250101, China; ymsun@sdjzu.edu.cn (Y.S.); dudunuo@163.com (C.C.); chinajnzx0026@163.com (C.D.); 2Jinan Eco Environmental Monitoring Center, Jinan 250101, China; liuguanghui2066@163.com

**Keywords:** PAHs, emission characteristics, particle size distribution, environmental impact, iron and steel enterprise

## Abstract

Download ash and emission dust samples were collected from sintering, coking, ironmaking and steelmaking processes of iron and steel enterprises in Laiwu. Sixteen kinds of polycyclic aromatic hydrocarbons (PAHs) in the United States Environmental Protection Agency (USEPA) priority controlled lists were quantitatively analyzed using Gas Chromatography-Mass Spectrometer (GC-MS). Laser particle size analyzer was used to obtain the distribution pattern of download ash. It was found that the diameter distribution pattern from four production processes was quite different. The proportion of fine particulate (0–2.5 μm) was the highest (72.62%) in the steelmaking refining process, and was 28.962% in the ironmaking process. Moreover, the particle size in download ash from steelmaking refining is all less than 10 μm and that from the ironmaking process was 52.92%. The medium-sized particles (10–100 μm) were dominant in sinter and coking download ashes. The total PAHs (∑_16_PAHs) mass concentration ranged from 0.49 ± 0.06 to 69.63 ± 5.57 μg·g^−1^ in download ash samples, and varied from 2.815 ± 0.253 to 19.429 ± 2.545 μg·m^−3^ in emission dust samples. The ∑_16_PAHs values were both largest in download ash and dust emission from the coking process (69.63 ± 5.57 μg·g^−1^ and 19.429 ± 2.545 μg·m^−3^, respectively). The most abundant individual PAHs were benzo[*b*]fluoranthene, benzo[*k*]fluoranthene, phenanthrene, benzo[*a*]anthracene in ash samples, and benzo[*a*]anthracene, benzo[*k*]fluoranthene, benzo[*b*]fluoranthene and indeno[1,2,3-*cd*]pyrene in emission dust samples. Dominant compounds were high-molecular weight (four- to six-ring) PAHs in both ash and dust samples. The concentration order of individual compounds in PM_10_ and PM_2.5_ in ambient air around the steel plant was completely consistent with each other, and the concentration of ∑_16_PAHs was the highest in the steel plant and lowest in Daqin village because of upwind of the steel plant. The concentrations of benzo[*b*]fluoranthene and fluoranthene in ambient air were comparatively high, and were in accordance with the higher concentration of the two monomers in the download ash samples, which suggested that the effect of the emission flue gas from the steel plant on ambient air was necessary to concern.

## 1. Introduction

Polycyclic aromatic hydrocarbons (PAHs) are a class of persistent organic pollutants widely existing in the environment with highly carcinogenic, mutagenic or teratogenicity effects [1,2]. Due to the harmful effects on human health, PAHs have been attracted much by scientists and scholars [3,4,5,6]. PAHs include hundreds of compounds, of which 16 PAHs have been recommended as priority pollutants by the United States Environmental Protection Agency (USEPA) in 1976 [7]. Numerous researches have revealed that sources of PAHs are complex and mostly came from human activities, such as incomplete combustion of fossil fuels or pyrolysis of fossil fuels during human activities [8,9,10]. PAHs generated through combustion are emitted to ambient air as particles and gases.

Iron and steel enterprises are an important source of PAHs, and the production processes of coking, sintering, ironmaking and steelmaking production activities, can produce a large number of PAHs which will seriously affect the ecological environment security in this region. Many scholars have conducted relevant studies on PAHs pollution and environmental hazards [11,12,13]. Gilio et al. [14] analyzed the source of PAHs through principal component analysis and positive matrix factorization, and the results proved the impact of iron and steel enterprises on the surrounding environment. Ooi et al. [15] analyzed the PAHs content in the four working environments of raw material inlet of sintering plant, sintering furnace row, coarse crushing roller and control room, indicating that the total PAHs content in the three sintering processes was higher than that in the control room. Wang et al. [16] analyzed the distribution regularity of PAHs and particle size of coking dust in five dust samples near the coking plant. Although there are many reports in the literature of the impacts of PAHs emitted during the sintering process on the surrounding environment, there are few reports on the effects caused by processes involved in coking, iron smelting, and steelmaking.

In this study, download ash and emission dust from different production processes of a certain iron and steel enterprise in Laiwu city were selected as the research objects. The concentration of 16 kinds of optimal control PAHs contained in it was studied to evaluate its potential risks. The research results can provide scientific basis for PAHs pollution control in the region where the iron and steel enterprise is located.

## 2. Results and Discussion

### 2.1. Particle Size Distribution of Download Ashes in Different Production Processes

It can be seen from Figure 1 that the high distribution peak of particle size of the download ash from the dust outfit of the sinter tank production processes appears at 100.65 μm. The particle size distribution in cooking is higher at 39.47 μm, where the distribution proportion is 8.25%. The high distribution peak of particle size in download ash from the taphole of the ironmaking process and steelmaking refining process appears in the range of fine particles (1.28 μm and 2.38 μm, respectively) and particle diameter is less than 2.5 μm. The distribution proportion for two different processes is 6.79% (taphole of ironmaking) and 22.13% (steelmaking refining), respectively. The distribution proportion download ash from the dust outfit of the steelmaking refining process is the highest, and the largest particle size is 8 μm, indicating that the fine particulate matter in this process ash is dominant. Fine dust particles and high dispersion have a greater impact on environmental pollution. However, the particle size distribution in download ash from the coking process appears in the 20–100 μm range. It suggests that large-sized particles are mainly in the dust outfit of this process. This result agrees with document research, that these particles mainly come from the mechanical processes such as crushing, screening and transfer of materials, and the proportion of large particles was relatively high [17]. Moreover, the particle size between the sinter tank and taphole of ironmaking processes are similar and much large-sized particles mainly appear in the dust outfit of two processes.

The particles which are less than or equal to 10 and 2.5 μm in aerodynamics equivalence diameter, respectively, are called inhaled particulate matter (PM) and represented as PM_10_ and PM_2.5_. PMs are an important component of atmospheric carbonaceous aerosols and PM_2.5_ is also called fine particles. Based on Table 1, the particle size in download ash from the dust outfit of steelmaking refining is all less than 10 μm, and the proportion of fine particulate (0–2.5μm) is the highest that reaches 72.62%. Additionally, the proportion of the small-sized particles (less than 10 μm) in download ash from the dust outfit of taphole of ironmaking was high at 52.92% (the sum of 29.86% and 28.06%). The middle-sized particles in download ash from the dust outfit of the coking process account for the vast majority (68.18%). The large-sized particles (100–300 μm) in download ash from the dust outfit of the four processes have a smaller proportion, although the value from the sinter tank process was comparatively high (20.97%). This indicates that inhaled particulate matter, especially fine particles, have a greater impact on environmental pollution.

### 2.2. Characteristics and Level of PAHs Pollution in Download Ash from Different Processes

Download ash concentration of PAHs in different production processes from iron and steel enterprises were given in Table 2. These data showed that the mass concentration of ∑_16_PAHs in the coking process is the highest, and the concentration of ∑_16_PAHs in the steelmaking process is the lowest, which is 0.49 ± 0.06 μg·g^−1^. BbF and BkF content are the highest in the coking process. Phe and Flua are the main components in the ironmaking process and Flua and BaA are the main components in the steelmaking process. PAHs in the coking process mainly come from the incomplete combustion of fossil fuels (mainly coal for the coking, while gas for sintering, iron smelting and steelmaking), and the processing process of tar, gas and other chemical products in each production workshop [18]. It is known that PAHs are mostly attached to small particles [19]. According to the particle size analysis, 10–100 μm particle size of download ash in the coking process is dominant, and the PAHs mass concentration is also the highest. The concentration of ∑_16_PAHs in the steelmaking plant is higher than that in the ironmaking plant, when there is little difference between the treatment process and the pollutant discharged. Environmentally relevant concentration of BaP significantly influenced the hepatic detoxification enzyme system and was one of the main cancer inducers. Table 3 also lists the proportion of BaP in 16 PAHs in download ash samples. BaP/∑_16_PAHs value is the highest in coking process ash samples. The proportion of BaP in 16 PAHs of ironmaking and steelmaking processes is less than 1.0%, which shows low environmental risk of two types of download ash.

### 2.3. Concentration Distribution of PAHs in Dust Emission from Different Processes

As can be seen from Table 3, except for Nap, Ace and Flu three compounds, other PAHs are detected in dust emission from four different processes and the total mass concentration of ∑_16_PAHs ranges from 2.815 ± 0.253 to 19.429 ± 2.545 μg·m^−3^. The highest concentration of ∑_16_PAHs appears in dust emission from the coking process (19.429 ± 2.545 μg·m^−3^), followed by the sintering process (4.435 ± 0.355 μg·m^−3^), and ∑_16_PAHs value in dust emission from steelmaking and ironmaking processes are roughly the same, respectively, 2.876 ± 0.377 μg·m^−3^ and 2.815 ± 0.253 μg·m^−3^. In sintering, steelmaking and ironmaking processes, single component BaA in dust emission was the highest, and the mass concentrations were 0.622 ± 0.055 μg·m^−3^, 0.428 ± 0.048 μg·m^−3^ and 0.366 ± 0.029 μg·m^−3^, respectively. The concentration of BkF in coking dust is the highest, which is 2.483 μg·m^−3^. The dust emission from the coking process is mainly generated in the process of coal loading, coke pushing, coke extinguishing and coke oven heating, which may also cause PAHs emission [20]. Zhu et al. [21] found that the mass concentrations difference in PAHs production processes was mainly caused by different combustion conditions. The combustion of coal in the coke oven was a dry distillation process, and severe hypoxia and high temperature in the furnace were conducive to the generation of PAHs, with the concentration of PAHs in smoke emission being relatively high. The download ash is the flue gas collected through the dust collector, and dust is the flue gas with organized emissions through the chimney after the cloth bag dust removal. Moreover, most of the small ring substances produced by pyrolysis are discharged into the atmosphere in the form of gas phase, and only a small part of them are attached to particles. In our collected samples, only the mass concentrations of Pha are higher than other small ring substances. This result indicates that the small ring substances of 16 PAHs are not dominating species in the dust emission from different processes. The sum of the individual PAHs in the four production processes is defined as ∑PAHs. It shows that the higher concentration individual PAHs are BaA, IcdP, BkF and BbF, with the value of 3.889 ± 0.311, 3.489 ± 0.314, 3.462 ± 0.318, 3.460 ± 0.311, respectively. The concentration of Bap, a heavily carcinogenic individual PAHs, was the highest in the coking process, with the value of 1.742 ± 0.156, which is consistent with the reported literature [20,22]. The proportion of Bap in 16 PAHs is similar and close to 8.80%. Compared with Table 2, the corresponding proportion in dusts is much higher than that in download ashes (maximum value was 6.89 ± 0.61). Hence, the emission flue gas should attract enough attention and it may have a serious impact on the surrounding atmosphere.

Figure 2 shows the distribution map of rings numbers of PAHs concentration in different processes dust emission. It indicates that the content of 5-ring is the highest in all processes dust, followed by 4-ring and 6-ring. Generally, low-molecular weight (2-ring and 3-ring) PAHs are mainly derived from oil leakage, while high-molecular weight (4-ring and above) PAHs are derived from incomplete combustion of fossil fuels such as coal [23,24].

### 2.4. Environmental Impact of PAHs in the Surrounding Air of the Steel Plant

In order to investigate the impact of the emission flue gas from the steel plant on the surrounding atmosphere, the detection of PAHs of particulate matter (PM) in ambient air around the steel plant is done and given in Figure 3. ∑16PAHs is the highest in the steel plant which is 3.579 μg·m^−3^ in PM_10_ and is 3.000 μg·m^−3^ in PM_2.5_, respectively. Documents reported the mass concentration of ∑_16_PAHs with 9.741 μg·m^−3^ in the environmental air at 1 km of a coking plant, using HPLC analysis method [25]. The ∑_16_PAHs concentration in our collected samples using GC-MS analysis method is lower than that in the literature. Seen from Figure 3, it can be found that the mass concentration of PAHs in PM_10_ and PM_2.5_ is the highest in the steel work. Daqin village is upwind of the plant, so its concentration of PAHs is the lowest. However, the concentration curve of three sample points in PM is similar to each other, suggesting that emission flue gas from the steel plant has certain influence on the surrounding environment. The highest mass concentration is BbF, and then it is Flua, in the three sampling sites with different particle sizes. In the Mengjiazhuang village sample, the mass concentration of BbF is 0.310 μg·m^−3^ and the mass concentration of Flua was 0.271 μg·m^−3^ in PM_10_. The values are lower than that the BbF concentration of atmospheric particulate matter in an urban area with 0.660 μg·m^−3^ in the whole year [26], because there is less industry in the urban area. However, our three sampling points are distributed around the steel plant area, so the concentration is significantly higher than that in the urban area. Compared with Table 2, the concentrations of BbF in ambient air were comparatively high. Therefore, the effect of the download ash from the steel plant on ambient air is necessary to concern.

## 3. Materials and Methods

### 3.1. Data and Sample Collection

A representative steel plant which is located in northern Laiwu, China, is selected as the research object. Emission dust from the four processes of sintering, coking, iron smelting and steelmaking was sampled three times every process with the flow rate 16.67 L/min by the diluting channel sampling equipment (made by Qingdao Laoshan Ltd., Qingdao, China), which was calibrated by a gas mass flow calibrator (API 700, New York, NY, USA). The sampling membrane diameter of quartz filter (Pallflex quartz filter membrane, New York, NY, USA) is 47 mm. Three samples were collected in each process, a total of 12 samples. At the same time, the download ash samples from the precipitator outfits of coking, steelmaking and ironmaking processes were collected directly three times every process. Each production process cleaned dusts every 3 h, and all download ashes were sampled for 10 min after one hour of dust cleaning.

According to the relevant requirements of the Technical Specifications for Environmental Air Quality Monitoring Points (Trial) (HJ664-2013), fully considering complex factors such as climate, geographical conditions and pollution sources near steel companies, two environmental sensitive sites were Mengjiazhuang and Daqinzhuang villages, located in the upwind and downwind of the plant, as well as a site inside the plant, were selected for monitoring PAHs in the environmental atmosphere. The sampling time was winter (25–30 December 2016). PM_2.5_ and PM_10_ filter samples were collected for 24 h using a median-flow particle sampler (Tianhong, Wuhan, Co. Ltd, Wuhan, China) with a flow rate of 100 L·min^−1^. Sampled quartz filters (φ90 mm) were used to analyze the PAHs.

### 3.2. Pretreatment and Determination of Samples

Before determination of samples, the download ashes of steel plant dust powder samples were pretreated by 105 μm steel-less mesh sieves and resuspended by a resuspending chamber, and then the quartz filter samples of PM_10_ were obtained. The filter samples were pretreated by Soxhlet extraction. The reagent n-hexane (residue analysis) was used as the extraction solvent (60 °C reflux frequency not less than 4 times per hour, the time of extraction was 16 h inside the Soxhlet apparatus). After evaporation to 2–3 mL with the rotary evaporator, the silica gel column was purified (leaching with mixed solution of *n*-hexane/dichloromethane (agricultural residues) with volume ratio of 1:1). After purification, nitrogen was blown to 0.5 mL with the nitrogen blower, which was transferred to the vial with *n*-hexane constant volume to 1 mL, and put in the refrigerator for testing.

Qualitative and quantitative analysis was performed by gas chromatography-mass spectrometry (GC-MS). The chromatographic column type is TG-5MS, quartz capillary column (30 m × 0.25 mm × 0.25 µm), injection temperature: 250 °C; column flow velocity: 1.10 mL/min; injection mode: split, split ratio is 10:1; column pressure: 69.3 kpa; and oven temperature: 40 °C. Sample quantity is 1.0 μL; ion source temperature: 200 °C; transfer line temperature: 250 °C; solvent switching time: 5 min; scanning mode adopts full scanning mode; and ion source is Electron impact ion (EI) source. Using programmed heating: Initial temperature: 70 °C, keeping 1 min, at 20 °C·min^−1^ up to 240 °C, with 10 °C·min^−1^ up to 310 °C and keeping 20 min. Carrier gas is pure helium.

The target compounds for monitoring and analysis were sixteen kinds of USEPA PAHs, and the specific substances are shown in Table 4.

### 3.3. Quality Control

In the blank samples, only Phe, Flua and Pyr were detected, while other PAHs compounds were not detected. The concentration of Phe, Flua and Pyr was far lower than the concentration in the actual samples, so it was negligible. It indicates that there is no interfering component of the target compound in the whole experiment [27]. At the time of detection, the standard solutions (2000 ug/mL, AccuStandard Inc., New York, NY, US) with concentrations of 0.2, 0.4, 0.6, 0.8, 1.0 and 2.0 mg·L^−1^ were configured, the correlation coefficient R^2^ were all above 0.9997. The recovery rate indicator pyrene is added to the blank sampling, and the marking recovery rate was between 82% and 113% (meeting the requirements of EPA70 ~130%), and the relative standard deviation was 1.74–12.6% (meeting the RSD < 20% specified by EPA) [28].

### 3.4. Particle Size Analysis Method

The laser particle size analyzer (LS-C (I), Zhuhai Oumeic Company, Zhuhai, China) was used to test the particle size distribution of download ashes. Before testing, the download steel plant dust powder samples were pretreated by 300 μm steel-less mesh sieve after the source of the sample size.

## 4. Conclusions

The diameter distribution patter in download ash from the dust outfit of four production processes was different. The proportion of fine particulate (0–2.5 μm) is the highest that reaches 72.62% in the steelmaking refining process. The medium-sized particles (10–100 μm) were dominant in the sinter tank and the coking process. Moreover, the proportion of the small-sized particles (less than 10 μm) in the ironmaking process was 52.92%.

The total levels of the sum of 16 USEPA priority PAHs (∑_16_PAHs) ranged from 0.49 ± 0.06 to 69.63 ± 5.57 μg·g^−1^ in download ash samples, and varied from 2.815 ± 0.253 to 19.429 ± 2.545 μg·m^−3^ in emission dust samples. The ∑_16_PAHs values in the coking process is the highest (69.63 ± 5.57 μg·g^-1^), and is the lowest in the steelmaking process, which is 0.49 ± 0.06 μg·g^−1^. The ∑_16_PAHs values was still biggest in dust emission from the coking process (19.429 ± 2.545 μg·m^−3^). The most abundant individual PAHs were BbF, BkF, Phe, BaA and Flua in ash samples, and BaA, BkF, BbF and IcdP in dust samples. Dominant compounds were 5-ring PAHs, which accounted for 41.0%, and 4-ring PAHs were about 30.0% in emission dusts. Although the proportion of BaP in 16 PAHs in ironmaking and steelmaking process ash samples was less than 1.0%, the proportion of Bap in 16 PAHs of the four processes was similar and close to 8.80% in dust emission.

The concentration curve of three surrounding atmosphere samples in PM_10_ and PM_2.5_ was similar to each other, and the mass concentration of ∑_16_PAHs was highest in the steel plant and lowest in Daqin village because of upwind of the steel plant. Compared with detected values in dust samples, the concentrations of BbF and Flua in ambient air were comparatively high. This result was consistent with the higher concentration of the two monomers in the download ash samples.

## Figures and Tables

**Figure 1 molecules-24-03646-f001:**
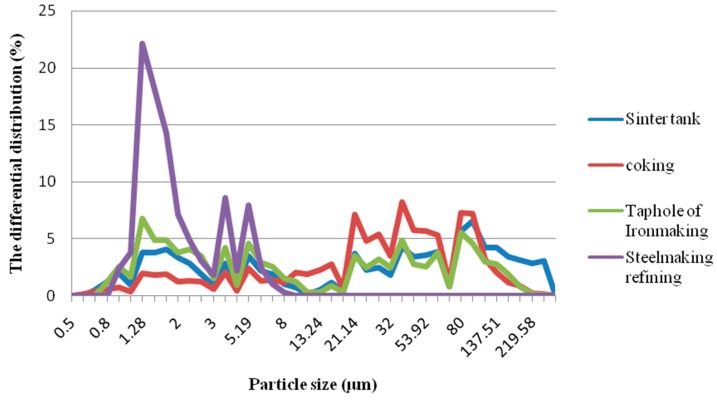
Distribution of particle size in different production processes.

**Figure 2 molecules-24-03646-f002:**
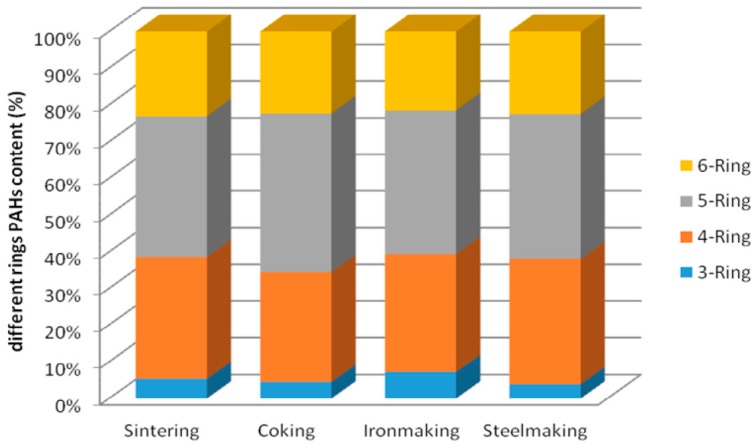
The distribution pattern of rings numbers of PAHs concentration in different processes dust emission.

**Figure 3 molecules-24-03646-f003:**
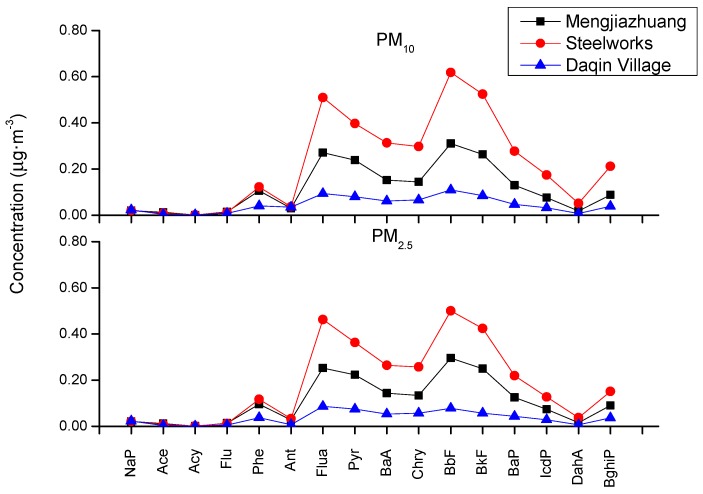
PAHs mass concentration in ambient air particulate matter around the steel plant.

**Table 1 molecules-24-03646-t001:** The different diameter distribution in download ash from dust outfit (%).

Diameter	Sinter Tank	Coking	Taphole of Ironmaking	Steelmaking Refining
0–2.5 μm	22.57	10.48	29.86	72.62
2.5–10 μm	15.70	12.71	23.06	27.38
10–100 μm	40.76	68.18	38.38	0
100–300 μm	20.97	7.63	8.70	0

**Table 2 molecules-24-03646-t002:** 16 PAHs mass concentrations (μg·g^−1^) in download ash from different processes.

	Coking	Ironmaking	Steelmaking
NaP	0.41 ± 0.03	0.30 ± 0.03	N.D.
Ace	0.43 ± 0.05	0.06 ± 0.02	N.D.
Acy	0.24 ± 0.03	0.18 ± 0.02	N.D.
Flu	1.43 ± 0.12	0.66 ± 0.06	N.D.
Phe	6.34 ± 0.05	2.68 ± 0.21	0.05 ± 0.01
Ant	2.03 ± 0.17	0.13 ± 0.03	0.01 ± 0.01
Flua	7.58 ± 0.68	1.40 ± 0.12	0.08 ± 0.03
Pyr	5.41 ± 0.65	0.49 ± 0.04	0.04 ± 0.02
BaA	5.63 ± 0.62	0.17 ± 0.02	0.08 ± 0.01
Chry	6.21 ± 0.75	0.18 ± 0.03	0.07 ± 0.02
BbF	9.80 ± 0.83	0.05 ± 0.02	0.04 ± 0.01
BkF	9.80 ± 1.17	0.05 ± 0.01	0.04 ± 0.02
BaP	4.80 ± 0.42	0.02 ± 0.01	0.02 ± 0.01
IcdP	3.10 ± 0.40	0.03 ± 0.02	0.03 ± 0.01
DahA	1.75 ± 0.21	0.02 ± 0.01	0.02 ± 0.01
BghiP	4.67 ± 0.42	0.03 ± 0.02	0.02 ± 0.01
∑_16_PAHs	69.63 ± 5.57	6.49 ± 0.62	0.49 ± 0.06
BaP/∑_16_PAHs (%)	6.89 ± 0.61	0.31 ± 0.03	0.99 ± 0.09

N.D. means none detected.

**Table 3 molecules-24-03646-t003:** PAHs mass concentration in dust emission from four different processes (μg·m^−3^).

	Sintering	Coking	Ironmaking	Steelmaking	∑PAHs
NaP	N.D.	N.D.	N.D.	N.D.	N.D.
Ace	N.D.	N.D.	N.D.	N.D.	N.D.
Acy	0.006 ± 0.001	N.D.	N.D.	N.D.	0.006
Flu	N.D.	N.D.	N.D.	N.D.	N.D.
Phe	0.148 ± 0.155	0.603 ± 0.054	0.149 ± 0.042	0.065 ± 0.008	0.966 ± 0.077
Ant	0.083 ± 0.010	0.280 ± 0.031	0.054 ± 0.007	0.047 ± 0.005	0.465 ± 0.042
Flua	0.137 ± 0.018	0.503 ± 0.040	0.111 ± 0.009	0.099 ± 0.008	0.852 ± 0.094
Pyr	0.263 ± 0.034	0.963 ± 0.125	0.166 ± 0.023	0.168 ± 0.022	1.561 ± 0.142
BaA	0.622 ± 0.055	2.472 ± 0.222	0.366 ± 0.029	0.428 ± 0.048	3.889 ± 0.311
Chry	0.451 ± 0.059	1.869 ± 0.241	0.260 ± 0.032	0.286 ± 0.022	2.868 ± 0.261
BbF	0.407 ± 0.036	2.482 ± 0.298	0.289 ± 0.023	0.280 ± 0.029	3.460 ± 0.311
BkF	0.407 ± 0.053	2.483 ± 0.276	0.289 ± 0.038	0.281 ± 0.023	3.462 ± 0.318
BaP	0.390 ± 0.046	1.742 ± 0.156	0.240 ± 0.032	0.258 ± 0.033	2.631 ± 0.292
IcdP	0.556 ± 0.072	2.244 ± 0.179	0.332 ± 0.049	0.356 ± 0.004	3.489 ± 0.314
DahA	0.495 ± 0.058	1.701 ± 0.189	0.283 ± 0.022	0.317 ± 0.042	2.797 ± 0.223
BghiP	0.465 ± 0.051	2.081 ± 0.169	0.271 ± 0.030	0.287 ± 0.025	3.105 ± 0.292
∑_16_PAHs	4.435 ± 0.355	19.429 ± 2.545	2.815 ± 0.253	2.876 ± 0.377	29.556 ± 3.281
BaP/∑_16_PAHs (%)	8.81 ± 1.13	8.97 ± 0.83	8.53 ± 0.68	8.97 ± 1.15	8.90 ± 0.71

N.D. means none detected.

**Table 4 molecules-24-03646-t004:** 16 USEPA polycyclic aromatic hydrocarbons (PAHs) names and properties.

Serial Number	Name	Abbreviations	Number of Benzene Ring	Limit of Detection (μg/mL)	Limit of Quantification (μg/mL)
1	naphthalene	NaP	2	0.002	0.005
2	acenaphthene	Ace	3	0.004	0.010
3	acenaphthylene	Acy	3	0.005	0.012
4	fluorine	Flu	3	0.005	0.013
5	phenanthrene	Phe	3	0.003	0.008
6	anthracene	Ant	3	0.004	0.011
7	fluoranthene	Flua	4	0.007	0.124
8	pyrene	Pyr	4	0.004	0.013
9	benzo[*a*]anthracene	BaA	4	0.002	0.007
10	chrysene	Chry	4	0.007	0.013
11	benzo[*b*]fluoranthene	BbF	5	0.003	0.006
12	benzo[*k*]fluoranthene	BkF	5	0.004	0.013
13	benzo[*a*]pyrene	BaP	5	0.004	0.007
14	indeno[1,2,3-*cd*] pyrene	IcdP	6	0.005	0.015
15	dibenz[*ah*]anthracene	DahA	5	0.006	0.013
16	benzo[*ghi*]perylene	BghiP	6	0.004	0.013

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
