# Peer review of "Distribution Pattern, Emission Characteristics and Environmental Impact of Polycyclic Aromatic Hydrocarbons (PAHs) in Download Ash and Dust from Iron and Steel Enterprise"

_molecules, 2019, doi:10.3390/molecules24203646_

Round 1

Reviewer 1 Report

This work compared PAH concentrations in particulates from different sources and atmospheric particulates. The manuscript has several interesting results. However, there are several points which are unclear or need more consideration. They are described below.

Major comments

For emission dust, three samples were collected from each process (Lines 60-63). However, standard deviations are not shown in Table 3. On the other hand, ash samples were collected (Lines 64-65). However, sample numbers are not described in the text and standard deviations are not shown in Table 4. Individual differences are very important for the reliability of analytical results. The dust sampling time was winter (Lines 63-64). This is not enough for the sampling condition, because the temperature is much higher at just after exhausting pipe than the atmosphere. The sampling temperature, affecting on the particle-gas distribution of PAHs, might be defined. AS for the sampling of ash, how long was the duration time between cleaning of precipitator and sampling? As described above the different sampling temperature may be considered as a reason for the different PAH concentration profiles between Tables 3 and 4. More discussion is necessary for the difference. “Compared with detected values in dust samples, the concentrations of BbP and Flua in ambient air were comparatively high. This result was consistent with the higher concentration of the two monomers in emission dust samples” was described in the conclusion (Lines 223-225). This is not correct, because the relative concentration of Flua in dust samples were not so high as in the atmosphere in Fig. 3. This manuscript wants to emphasize the characteristics of PAH emission from iron and steel enterprise. However, samples were collected stational sources such as emission processes using coal only and it is well known that automobiles are major stational contributor of PAHs. In order to describe the contribution of iron and steel enterprise to atmospheric PAHs, contributions of automobiles might be necessary.

Minor comments

Typographical errors in Abstract, as examplesbenzo[b]fluoranthene → benzo[b]fluoranthene (italic)benzo[k]-fluoranthene → benzo[k]fluoranthene (delete “-“ and italic); indeno [123-cd] pyrene→ indeno[1,2,3-cd]pyrene (delete space, add “,” and italic); Check others in the text.

Author Response

Reviewer 1

Comments and Suggestions for Authors.

This work compared PAH concentrations in particulates from different sources and atmospheric particulates. The manuscript has several interesting results. However, there are several points which are unclear or need more consideration. They are described below.

Major comments

For emission dust, three samples were collected from each process (Lines 60-63). However, standard deviations are not shown in Table 3.

Response: Thank you very much for the suggestion. The standard deviations were showed in Table 3. 

On the other hand, ash samples were collected (Lines 64-65). However, sample numbers are not described in the text and standard deviations are not shown in Table 4. Individual differences are very important for the reliability of analytical results.

Response: The standard deviations were showed in Table 4.

The dust sampling time was winter (Lines 63-64). This is not enough for the sampling condition, because the temperature is much higher at just after exhausting pipe than the atmosphere. The sampling temperature, affecting on the particle-gas distribution of PAHs, might be defined.

Response: dilution channel sampling equipment was used to collect the dust samples. This sampling equipment was simulated to make it fully meet the atmospheric environment. Therefore, the temperature difference between the sampling port and the ambient air had little effect on the particle distribution of PAHs.

AS for the sampling of ash, how long was the duration time between cleaning of precipitator and sampling?

Response: The duration time between cleaning of precipitator and sampling was added in the revised version. Each production processes cleaned dusts every three hours, and all download ash were sampled for 10 minutes after one hour of dust cleaning.

As described above the different sampling temperature may be considered as a reason for the different PAH concentration profiles between Tables 3 and 4.

Response: Because the dust emission samples and the dust download ash were collected in the same temperature from the same process. For the ash, we directly sampled ash from the outlet of the precipitator and for the aerosol PAHs in the flow dust. dilution channel sampling equipment was used to collect the dust emission samples in the flow gas for the different processes and was simulated to make it fully meet the atmospheric simulation experiment.

More discussion is necessary for the difference. “Compared with detected values in dust samples, the concentrations of BbFand Flua in ambient air were comparatively high. This result was consistent with the higher concentration of the two monomers in emission dust samples” was described in the conclusion (Lines 223-225). This is not correct, because the relative concentration of Flua in dust samples were not so high as in the atmosphere in Fig. 3.

Response: we have carefully analyzed the data in Fig. 3 and Table 3 and changed this sentence into “BbF and Flua in ambient air were comparatively high. This result was consistent with the higher concentration of the two monomers in coking download ash samples(see Table3) 

This manuscript wants to emphasize the characteristics of PAH emission from iron and steel enterprise. However, samples were collected stational sources such as emission processes using coal only and it is well known that automobiles are major stational contributor of PAHs. In order to describe the contribution of iron and steel enterprise to atmospheric PAHs, contributions of automobiles might be necessary.

Response: Such is the case that automobiles are also stational contributor of PAHs in ambient air. PM2.5 and PM10 in this study were sampled from two environmental sensitive sites that located in the village in Laiwu city. This town is less urbanized and slow-developing city and the number of automobiles is not large. Because the sampling time was in winter, agricultural vehicles were not often used. The literatures reported that the PAHs emission of motor vehicles has distinct seasonal variation, and the contribution of automobiles emission is obviously lower in winter [1]. So the impact of automobiles emissions on PAHs was not considered in this study.

[1] Fei C, Wei H, Qin Z. Emissions of particle-phase polycyclic aromatic hydrocarbons (PAHs) in the Fu Gui-shan Tunnel of Nanjing, China[J]. Atmospheric Research, 2013, 124(124):53-60.

Minor comments

Typographical errors in Abstract, as examplesbenzo[b]fluoranthene → benzo[b]fluoranthene (italic)benzo[k]-fluoranthene → benzo[k]fluoranthene (delete “-“ and italic); indeno [123-cd] pyrene→ indeno[1,2,3-cd]pyrene (delete space, add “,” and italic); Check others in the text. 

Response: All typographical errors were corrected in the revised version.

Reviewer 2 Report

This paper is interesting and provides new data of PAH source from iron and steel processes. Nevertheless is important to clarify some issues-

Abstract

Page 1. Line 13-14

Please rewrite the paragraph. “Was highest that reached 100%” has no sense. I thin highest should be eliminated.

Materials and methods

In this section there is no mention about the collected samples in the plant and in the surroundings. Where is the town? There is not either explanation of how were collected PM2.5 and PM10. Which kind of samplers, how many days and hours?. Authors should be specific about this.

How was calibrated the dilution chamber. This is very important.

Authors mentioned several times an EPA reference but they did not specify which method was used. There is more than one. Include also in the reference list.

I do not think that nitrogen was the carrier gas. All GC-MS methods for PAH use pure helium. Please explain.

Which kind of PAH standards were used for standard curves?

How were determined PAHs recoveries? You should explain this issue in detailed form, since you only use PAH standards. Please include in Table 1 the recovery of each compound.

Results

Page 4. Line 125-128

The paragraph is confused. If PM2.5 in reaches 72.62%, Based on the Table 2, the particle size in download ash from dust outfit of steelmaking 126 refining is all less than 10μm and the proportion of PM2.5 is highest that reaches 72.62%. Also, the 127 proportion of the small-sized particles (less than 10 μm) in download ash from dust outfit of 128 taphole of ironmaking was high that was 52.92%.

Page 5

Authors claimed in Lines 141-142 that “PAHs in the coking process mainly comes from the incomplete combustion of  fossil fuels and the processing process of tar, gas and other chemical products in each production  workshop [15]”. Then it is quite important to report which are the fossil fuels used in the different process?

Line 144. Again the term highest is not well used. Please review the English. Highest should be used as “the highest”.

Page 6. Line 164

What do you mean that “some parts are not strict or poorly operated”. How do that can contribute to the high PAH concentrations? For instance, I do not think that coal loading can contribute to PAH concentrations since there is not combustion.

How do you explain that the BAP proportion in dust be very different that in particle concentration.

Authors talk about air quality data collected by them and compared with some literature. But it is important to explain the differences among the sampling made by the authors and in other studies. Equipment and methodologies should be compared.

Which other PAH sources are in the village?

Please improve conclusions after corrections

Author Response

Reviewer 2

Comments and Suggestions for Authors.

This paper is interesting and provides new data of PAH source from iron and steel processes. Nevertheless is important to clarify some issues-

Abstract

Page 1. Line 13-14.Please rewrite the paragraph. “Was highest that reached 100%” has no sense. I think highest should be eliminated.

Response: Thank you very much for the comments. This paragraph was rewrited and the sentence “Was highest that reached 100%” had been eliminated in the revised version.

Materials and methods

1) In this section there is no mention about the collected samples in the plant and in the surroundings. Where is the town? There is not either explanation of how were collected PM2.5 and PM10. Which kind of samplers, how many days and hours?. Authors should be specific about this.

Response: The details of sampling have been added in paragraph 2.1. A representative steel plant which locates in northern Laiwu, China, is selected as the research object. Emission dust from the four processes of sintering, coking, iron smelting and steelmaking was sampled three times every process with the flow rate 16.67L/min by the diluting channel sampling equipment (Made by Qingdao laoshan Ltd.), which was calibrated by a gas mass flow calibrator (API 700, USA) . The sampling membrane diameter of quartz filter (Pallflex quartz filter membrane, USA) is 47 mm. Three samples were collected in each process, a total of 12 samples. At the same time, the download ash samples from the precipitator outfits of coking, steelmaking and ironmaking processes were collected directly three times every process. Each production processes cleaned dusts every three hours, and all download ash were sampled for 10 minutes after one hour of dust cleaning.

According to the relevant requirements of the Technical Specifications for Environmental Air Quality Monitoring Points (Trial) (HJ664-2013), fully considering complex factors such as climate, geographical conditions and pollution sources near steel companies, two environmental sensitive sites were Mengjiazhuang and Daqinzhuang villages, located in the upwind and downwind of the plant, as well as a site inside the plant, were selected for the monitoring of PAHs in the environmental atmosphere. The sampling time was winter (2016.12.25-12.30). PM2.5 and PM10 filter samples were collected for 24 hours using a median-flow particle samplers (Tianhong, Wuhan, Co. Ltd) with a flow rate of 100 L·min−1. Sampled quartz filters (φ90mm) were used to analyze the PAHs.

2) How was calibrated the dilution chamber. This is very important.

Response: We have explained how to calibrate the dilution chamber in line 65. Emission dust from the four processes of sintering, coking, iron smelting and steelmaking was sampled three times every process with the flow rate 16.67L/min by the diluting channel sampling equipment (Made by Qingdao laoshan Ltd.), which was calibrated by a gas mass flow calibrator (API 700, USA) .

3) Authors mentioned several times an EPA reference but they did not specify which method was used. There is more than one. Include also in the reference list.

Response: The reference of EPA as reference14 has been added in the revised article. 

4) I do not think that nitrogen was the carrier gas. All use pure helium. Please explain.

Response: It was very sorry that we made an error. In fact, the carrier gas for PAHs with GC-MS methods was pure helium. We have corrected in the line 98 in the revised version.

5) Which kind of PAH standards were used for standard curves?

Response: The standard solutions (2000ug/ml, Accustandard Inc. US) with concentrations of 0.2, 0.4, 0.6, 0.8, 1 and 2 mg·L-1 were configured, the correlation coefficient R2 were all above 0.9997.

6) How were determined PAHs recoveries? You should explain this issue in detailed form, since you only use PAH standards. Please include in Table 1 the recovery of each compound.

Response: The recovery rate indicator pyrene is added to the blank sampling, and the marking recovery rate was between 82% ~ 113% (meeting the requirements of EPA70 ~ 130%), and the relative standard deviation was 1.74% ~ 12.6% (meeting the RSD<20% specified by EPA).

Results

1) Page 4. Line 125-128. The paragraph is confused. If PM2.5 in reaches 72.62%, Based on the Table 2, the particle size in download ash from dust outfit of steelmaking 126 refining is all less than 10μm and the proportion of PM2.5 is highest that reaches 72.62%. Also, the 127 proportion of the small-sized particles (less than 10 μm) in download ash from dust outfit of 128 taphole of ironmaking was high that was 52.92%.

Response: The paragraph was revised in Page 4 line 125-128 of original paper. Based on the Table 2, the particle size in download ash from dust outfit of steelmaking refining is all less than 10 μm and the proportion of fine particulate (0-2.5μm) is the highest that reaches 72.62%. Also, the proportion of the small-sized particles (less than 10 μm) in download ash from dust outfit of taphole of ironmaking was high that was 52.92%(the sum of 29.86% and 28.06% in the revised version.

2) Page 5: Authors claimed in Lines 141-142 that “PAHs in the coking process mainly comes from the incomplete combustion of fossil fuels and the processing process of tar, gas and other chemical products in each production workshop [15]”. Then it is quite important to report which are the fossil fuels used in the different process?

Response: we added the fuels in Page 5 in lines 159-160: mainly coal for the coking, while gas for sintering, iron smelting and steelmaking.

3) Line 144. Again the term highest is not well used. Please review the English. Highest should be used as “the highest”.

Response: We checked carefully the whole paper and “the highest” was used in the revised version.

4) Page 6. Line 164. What do you mean that “some parts are not strict or poorly operated”. How do that can contribute to the high PAH concentrations? For instance, I do not think that coal loading can contribute to PAH concentrations since there is not combustion.

Response: We have deleted the sentence “some parts are not strict or poorly operated” in the revised version.

5) How do you explain that the BAP proportion in dust be very different that in particle concentration.

Response: The Bap proportion in dust samples of four different processes (Table 4) was similar and approximately 8.80%.

6) Authors talk about air quality data collected by them and compared with some literature. But it is important to explain the differences among the sampling made by the authors and in other studies. Equipment and methodologies should be compared.

Response: The differences between our sampling and the reference were added in line 226and 227.

7) Which other PAH sources are in the village?

Response: Such is the case that automobiles are also stational contributor of PAHs in ambient air. PM2.5 and PM10 in this study were sampled from two environmental sensitive sites that located in the village in Laiwu city. This town is less urbanized and slow-developing city and the number of automobiles is not large. Because the sampling time was in winter, agricultural vehicles were not often used. The literatures reported that the PAHs emission of motor vehicles has distinct seasonal variation, and the contribution of automobiles emission is obviously lower in winter [1]. So the impact of automobiles emissions on PAHs was not considered in this study.

[1] Fei C, Wei H, Qin Z. Emissions of particle-phase polycyclic aromatic hydrocarbons (PAHs) in the Fu Gui-shan Tunnel of Nanjing, China[J]. Atmospheric Research, 2013, 124(124):53-60.

8) Please improve conclusions after corrections

Response: The conclusions have been improved according to the reviewer’s suggestion.

Reviewer 3 Report

GENERAL COMMENT

Authors described the particle size distribution of the ash collected from the four different processes (sintering, coking, ironmaking and steel making). Simultaneously, the concentration of 16PAHs was determined in the ash and emitted dust. However, only 12 samples were collected (3 samples for each process) to characterize the PAHs and dust emission. Formulating conclusions based on such small amount of samples is unjustified. In addition, the objective of the study is of low novelty, there is a lack of any statistical evaluation of obtained results, therefore, the manuscript  does not meet criteria of today’s science.

The way of preparing the manuscript also raises many objections. The manuscript is poorly written, there are many unclear sentences. Someone familiar with English as used in the scientific literature should copyedit the text of the paper.

SPECIFIC COMMENTS

As a general comment: authors have cited rather old literature, a lot of items are from the 90ties; additionally, in the text authors refer to incorrect publications of other authors (e.g. line 33 and 36) thus all citations in the manuscript need to be checked.

Line 18-19: please correct the name of PAH compounds (here and in the whole manuscript), it should be benzo[k]fluoranthene, benzo[a]anthracene

Line 24-27: the following sentence is unclear, please correct: “The concentrations of benzo[b]fluoranthene and fluoranthene in ambient air were comparatively high, and in accordance with the higher concentration of the two monomers in the emission dust samples, which suggested that the effect of the emission flue gas from the steel plant on ambient air was necessary to concern.”

Line 43: it should be Gilio et al. [10]…

Line 46: it should be Ooi et al. [11]…

Line 49: it should be Wang et al. …; what does it mean “the distribution law of PAHs”?, please clarify.

Line 54-55: what does it mean “…the condition of 16kinds of optimal control PAHs contained…” please clarify.

Materials and Methods section

It should be clearly stated what was studied and what kind of methods was used.

The description of samples preparation for PAHs analysis and GS-MS determination of hydrocarbons is chaotic and unclear in many points.

Lines 70-72: the sentence is not clear, please clarify

Line 80: what does it mean “shunt, shunt ratio is 10:1”? Explain “Column temperature” .. this is oven temperature? If yes, why 40°C?

Line 81: “transmission temperature”? or transfer line temperature?

Line 84: usually in the GS-MS analysis of PAHs helium is used as carrier gas. Please check..

Line 91: cited reference [13] is referred to PCBs not PAHs analysis, please provide proper citation.

Lines 92-95: as quality control parameters limit of detection and limit of quantification should be given for individual PAHs compounds.

Table 1: please correct the names of PAH compounds – it should be fluorene, benzo[k]fluoranthene, benzo[a]anthracene

Results and discussion

All results should be evaluated using statistical methods,  the results without indicating their significance are worthless. Please provide statistical analysis and discuss the results again!!

Line 123, 132, 189: it should be “particulate matter (PM)”

Lines 135-151: please check results reported in this paragraph because in they are not consistent with data presented in Table 3!!

Table 3: please provide statistical analysis of the results; what 0.00 values mean? PAH compound is not detected or is under detection limit? please clarify

Line 164: it should be Zhu et al. [18]…

Table 4: the statistical analysis should be included; to what refer ΣPAHs in the last column? Please explain and clarify

Line 181-182: it should be “the distribution pattern”..

Lines 187-205: the analysis of the PAHs in ambient air is not mentioned in the Materials and method section.  Please clarify this. 

Conclusions

This section should be rewritten after proper statistical analysis of the obtained results!!

Author Response

Reviewer 3

Comments and Suggestions for Authors

GENERAL COMMENT

Authors described the particle size distribution of the ash collected from the four different processes (sintering, coking, ironmaking and steel making). Simultaneously, the concentration of 16PAHs was determined in the ash and emitted dust. However, only 12 samples were collected (3 samples for each process) to characterize the PAHs and dust emission. Formulating conclusions based on such small amount of samples is unjustified. In addition, the objective of the study is of low novelty, there is a lack of any statistical evaluation of obtained results, therefore, the manuscript  does not meet criteria of today’s science. The way of preparing the manuscript also raises many objections. The manuscript is poorly written, there are many unclear sentences. Someone familiar with English as used in the scientific literature should copyedit the text of the paper.

SPECIFIC COMMENTS

1) As a general comment: authors have cited rather old literature, a lot of items are from the 90ties; additionally, in the text authors refer to incorrect publications of other authors (e.g. line 33 and 36) thus all citations in the manuscript need to be checked.

Response: Thank you very much for the comments. The 90ties literatures were renewed and the incorrect publications of other authors have been checked in the revised version.

2) Line 18-19: please correct the name of PAH compounds (here and in the whole manuscript), it should be benzo[k]fluoranthene, benzo[a]anthracene

Response: The name of PAH compounds have been checked and corrected.

3) Line 24-27: the following sentence is unclear, please correct: “The concentrations of benzo[b]fluoranthene and fluoranthene in ambient air were comparatively high, and in accordance with the higher concentration of the two monomers in the emission dust samples, which suggested that the effect of the emission flue gas from the steel plant on ambient air was necessary to concern.”

Response: The sentences were revised “The concentrations of benzo[b]fluoranthene and fluoranthene in ambient air were comparatively high, and in accordance with the higher concentration of the two monomers in the download ash samples, which suggested that the effect of the emission flue gas from the steel plant on ambient air was necessary to concern” 

 4)Line 43: it should be Gilio et al. [10]…

Response: According to the suggestion, we have corrected the sentence.

5) Line 46: it should be Ooi et al. [11]…

Response: The sentence has been corrected in the revised version.

6) Line 49: it should be Wang et al. …; what does it mean “the distribution law of PAHs”?, please clarify.

Response: We read again the reference by Wang et al and have corrected the sentence“the distribution regularity of PAHs”.

7)Line 54-55: what does it mean “…the condition of 16kinds of optimal control PAHs contained…” please clarify.

Response: The sentence“…the condition of 16kinds of optimal control PAHs contained…” has changed into “…the concentration of 16kinds of optimal control PAHs contained…”.

Materials and Methods section

8) It should be clearly stated what was studied and what kind of methods was used.

Response: We have clearly stated it in section 2.1 including the kind of methods, samplers, date and town et. al.

9) The description of samples preparation for PAHs analysis and GS-MS determination of hydrocarbons is chaotic and unclear in many points.

Response: pretreatment and determination of samples for PAHs were revised according to the reviewer suggestions and seen in section 2.2.

10) Lines 70-72: the sentence is not clear, please clarify

Response: The lines 70-72 sentence in the original paper were changed intoThe reagent n-hexane ( residue analysis) was used as the extraction solvent (60℃, reflux frequency not less than 4 times per hour, the time of extraction 16 hours inside the soxhlet apparatus ). 

11) Line 80: what does it mean “shunt, shunt ratio is 10:1”? Explain “Column temperature” .. this is oven temperature? If yes, why 40°C?

Response: The word shunt has revised split. The “Column temperature” should be “oven temperature” in the revised version.

12) Line 81: “transmission temperature”? or transfer line temperature?

Response: It should be transfer line temperature in the revised version.

13) Line 84: usually in the GS-MS analysis of PAHs helium is used as carrier gas. Please check.

Response: It was very sorry that we made an error. In fact, the carrier gas for PAHs with GC-MS methods was pure helium. We have corrected in the revised version..

14) Line 91: cited reference [13] is referred to PCBs not PAHs analysis, please provide proper citation.

Response: The reference 13 was changed in the revised version.

15) Lines 92-95: as quality control parameters limit of detection and limit of quantification should be given for individual PAHs compounds.

Response: The limit of detection and limit of quantification have been added in Table1.

16) Table 1: please correct the names of PAH compounds – it should be fluorene, benzo[k]fluoranthene, benzo[a]anthracene

Response: All typographical errors were corrected in the revised version.

Results and discussion

17) All results should be evaluated using statistical methods, the results without indicating their significance are worthless. Please provide statistical analysis and discuss the results again!!

Response: The standard deviations have been added in Table 3 and Table 4.

18) Line 123, 132, 189: it should be “particulate matter (PM)”

Response: The names of particulate matter (PM) were checked and all corrected in the revised version.

19) Lines 135-151: please check results reported in this paragraph because in they are not consistent with data presented in Table 3!!

Response: The data in Table 3 were carefully analyzed and results were checked in section 3.3.

20) Table 3: please provide statistical analysis of the results; what 0.00 values mean? PAH compound is not detected or is under detection limit? please clarify

Response: The standard deviations in Table 3 and Table 4 were given and the 0.00 values means not detected. 0.00 values” were changed into N. D.

21) Line 164: it should be Zhu et al. [18]…

Response: The sentence was corrected in the revised version.

22) Table 4: the statistical analysis should be included

Response: The statistical analysis was included in Table 4.

23) to what refer ΣPAHs in the last column? Please explain and clarify

Response: The ΣPAHs refer the sum of the individual PAHs in the four production process. The explained was as follows. The sum of the individual PAHs in the four production process is defined as ∑PAHs, it shows that the higher concentration individual PAHs is BaA, IcdP, BkF and BbF, with the value of 3.889±0.311, 3.489±0.314, 3.462±0.318, 3.460±0.311, respectively. The concentration of Bap, a heavily carcinogenic individual PAHs, was the highest in cooking process, with the value of 1.742±0.156, which is consistent with the reported literature [22, 24].

24) Line 181-182: it should be “the distribution pattern”..

Response: “The distribution pattern” was corrected in the revised version.

25)Lines 187-205: the analysis of the PAHs in ambient air is not mentioned in the Materials and method section.  Please clarify this. 

Response: The details sampling method and analysis of the PAHs in ambient air were added in paragraph 2.1. According to the relevant requirements of the Technical Specifications for Environmental Air Quality Monitoring Points (Trial) (HJ664-2013), fully considering complex factors such as climate, geographical conditions and pollution sources near steel companies, two environmental sensitive sites were Mengjiazhuang and Daqinzhuang villages, located in the upwind and downwind of the plant, as well as a site inside the plant, were selected for monitoring PAHs in the environmental atmosphere. The sampling time was winter (2016.12.25-12.30). PM2.5 and PM10 filter samples were collected for 24 hours using a median-flow particle samplers (Tianhong, Wuhan, Co. Ltd) with a flow rate of 100 L·min−1. Sampled quartz filters (φ90mm) were used to analyze the PAHs.

Conclusions

26) This section should be rewritten after proper statistical analysis of the obtained results!!

Response: The conclusions have been improved according to the reviewer’s suggestion.

Round 2

Reviewer 1 Report

According to reviewer's comments, the author has revised the manuscript, which is acceptable for the publication.

Reviewer 2 Report

The manuscript has been improved

Reviewer 3 Report

I have had a lot of comments and recommendations previously, authors have addressed all of them. Many required corrections have been made in the manuscript, however the authors have not avoided mistakes.

I have two remarks:

The paper should be carefully checked for typos and lack of spaces. The method of writing the units should be checked and corrected if necessary. The authors must decide on the presentation of units, e.g. mg/L or mg·L-1, ml/min or ml·min-1.